# Cardiometabolic Risk Assessment in a Cohort of Children and Adolescents Diagnosed with Hyperinsulinemia

**DOI:** 10.3390/diseases12060119

**Published:** 2024-06-04

**Authors:** Giorgio Sodero, Donato Rigante, Lucia Celeste Pane, Linda Sessa, Ludovica Quarta, Marcello Candelli, Clelia Cipolla

**Affiliations:** 1Department of Life Sciences and Public Health, Fondazione Policlinico Universitario A. Gemelli IRCCS, 00168 Rome, Italy; donato.rigante@unicatt.it (D.R.); clelia.cipolla@policlinicogemelli.it (C.C.); 2Department of Pediatrics, Università Cattolica Sacro Cuore, 00168 Rome, Italy; 3Department of Clinical Internal, Anesthesiologic and Cardiovascular Sciences, Sapienza University of Rome, 00161 Rome, Italy; 4Department of Emergency Anesthesiological and Reanimation Sciences, Fondazione Policlinico Universitario A. Gemelli IRCCS, 00168 Rome, Italy

**Keywords:** hyperinsulinism, child, cardiovascular risk, pediatric endocrinology, personalized medicine

## Abstract

Background: Individuals with hyperinsulinemia may initially not meet any diagnostic criteria for metabolic syndrome, though displaying a higher risk of cardiovascular complications combined with obesity, diabetes, and hypertension. Aim: The main objective of our study was to assess the diagnostic accuracy of various cardiovascular risk indices in hyperinsulinemic children and adolescents; a secondary objective was to estimate the optimal cut-offs of these indices. Patients and methods: This retrospective single-center study was conducted on 139 patients aged 12.1 ± 2.9 years, managed for hyperinsulinism. Results: We found statistically significant differences in homeostasis model assessment of insulin resistance index (HOMA-IR), triglyceride glucose index (TyG), TyG-body mass index, visceral adiposity index, lipid accumulation product index, fatty liver index, and hepatic steatosis index. At the linear logistic regression assessment, we found that insulin growth factor-1 (IGF-1), HOMA-IR, and ALT/AST ratio were independently associated with confirmed hyperinsulinism. At the multivariate analysis, IGF-1 levels over 203 ng/mL and HOMA-IR higher than 6.2 were respectively associated with a 9- and 18-times higher odds ratio for hyperinsulinism. The other investigated parameters were not significantly related to hyperinsulinism, and could not predict either the presence of hyperinsulinemia or a subsequent cardiovascular risk in our patients. Conclusion: Commonly used indices of cardiovascular risk in adults cannot be considered accurate in confirming hyperinsulinism in children, with the exception of HOMA-IR. Further studies are needed to verify the usefulness of specific cardiovascular risk indices in hyperinsulinemic children and adolescents.

## 1. Introduction

Metabolic syndrome is defined by a cluster of cardiovascular risk factors associated with insulin resistance, visceral obesity, and unbridled systemic inflammation [1]. Although more common in adulthood, the metabolic syndrome has become a relevant cause of morbidity in the pediatric age, due to an increasing incidence of non-genetic obesity determined by both unbalanced lifestyles and incorrect dietary habits [2].

The etiology of obesity differs between adults and pediatric populations: in fact, childhood obesity is influenced by a complex interplay of genetic, environmental, and behavioral factors, including parental obesity, early feeding practices, and socio-economic status [3]. However, long-term adverse effects of obesity are shared between the two categories, and it has been demonstrated that the cardiovascular risk in the adult population is also associated with different metabolic features in early life [2].

According to the International Diabetes Federation [4], children aged 10–15 years meet the criteria for metabolic syndrome if they exhibit obesity with waist circumference (WC) > 90th percentile and two risk factors among blood pressure ≥ 130/85 mm_Hg_, density lipoprotein (HDL) cholesterol ≤ 40 mg/dL, triglycerides ≥ 150 mg/dL, or fasting glucose ≥ 100 mg/dL. Notwithstanding, it is not possible to diagnose this syndrome in children under 10 years due to the absence of age-specific reference values. Even if individuals with hyperinsulinemia may not initially meet the diagnostic criteria, it is known that they are at higher risk of developing metabolic syndrome, metabolic dysfunction-associated steatotic fatty liver disease, diabetes, hypertension, and further cardiovascular complications [1,4]. Cardiovascular diseases are currently the leading cause of mortality in the industrialized countries [1], and the rate of cardiovascular events may even start during the pediatric age [5], warranting the need to mitigate this risk with the aim of improving morbidities in adulthood and overall life expectancy.

It is also known that certain anthropometric parameters, alone or in combination with laboratory data, might identify groups of patients at higher risk of developing the metabolic syndrome [6]. Over time, cardiovascular risk indices have therefore been developed based on the combination of multiple auxological parameters, laboratory markers, or a combination of both. An alteration of these indirect indices has been statistically associated with both hyperinsulinemia and risk of developing metabolic syndrome [7]. However, the majority of studies pertains to adult patients, while there are limited data referred to the pediatric population. We performed this retrospective study to evaluate the applicability of various cardiovascular risk indices to the pediatric population diagnosed with hyperinsulinism.

## 2. Patients and Methods

This is a cross-sectional, retrospective, single-center study conducted at the Pediatric Endocrinology Day Hospital of the Fondazione Policlinico Universitario A. Gemelli IRCCS, Rome, in children evaluated for hyperinsulinism. The main objective of our study was to assess the diagnostic accuracy of various cardiovascular risk indices—commonly used in adulthood—for studying hyperinsulinemic pediatric patients; a secondary objective was to estimate the optimal cut-offs of these indices in our sample.

### 2.1. Inclusion and Exclusion Criteria

The inclusion criteria of our study were as follows:Age < 18 years.Suspected hyperinsulinism (defined as basal insulin > 25 OR family history of hyperinsulinism and/or diabetes OR presence of acanthosis nigricans and/or other signs of hyperinsulinism on the physical examination OR body mass index (BMI), waist circumference (WC), and/or hip circumference (HC) > 95th percentile according to age and sex).Complete endocrinological follow-up, including blood pressure measurement, abdominal ultrasound with assessment of hepatic parenchyma features, auxological parameters (weight, height, BMI, WC, HC; for each variable we calculated the respective standard deviation [SD] for age and sex), laboratory test (transaminases, complete thyroid profile, including thyroid-stimulating hormone (TSH), triiodothyronine (fT3), thyroxine (fT4), insulin growth factor-1 (IGF-1), insulin, glucose, glycated hemoglobin, uric acid, complete lipid profile including total cholesterol, triglycerides, low density lipoprotein (LDL), HDL, very low density lipoprotein (VLDL)), and oral glucose tolerance test (OGTT) results, including insulin and glucose peak.

The exclusion criteria of our study were as follows:Diagnosis of diabetes mellitus, defined as glycated hemoglobin ≥ 6.5% OR fasting plasma glucose ≥ 126 mg/dL OR glucose ≥ 200 mg/dL during an OGTT, OR random glucose ≥ 200 mg/dL in a patient with classic diabetic symptoms, like polyuria and/or polydipsia [7].Coded diagnosis of metabolic syndrome (obesity with waist circumference > 90th percentile and two risk factors among blood pressure ≥ 130/85 mm_Hg_, HDL ≤ 40 mg/dL, triglycerides ≥ 150 mg/dL, or fasting glucose ≥ 100 mg/dL [4].Therapy with metformin and/or other antidiabetic drugs (excluding patients who started a specific therapy following the execution of the OGTT).

After the screening process, we retrospectively collected information from the medical records of 139 children and young adolescents; we divided patients into two groups based on the result of the OGTT.

Currently, the OGTT represents the main tool for assessing insulin peak, both in pediatric and adult populations. In the absence of a standardized guideline for interpreting insulin levels during OGTT, there is significant variability in the cut-off values, which may depend on the type of guidelines considered or clinicians’ experience. Currently, a defined cut-off value in the pediatric population has not yet been established. However, several possible cut-offs exist, including [8,9] the following:✓ The sum of insulin measurements at different sampling times during the OGTT > or <2083.5 pmol/L (300 μU/mL).✓ An insulin peak ≥ 1041.75 pmol/L.✓ A blood insulin value ≥ 520.88 pmol/L (75 μU/mL) when sampled 120 min after glucose loading.✓ An insulin peak above 100 uIU/mL.

In our center, in the absence of a commonly shared guideline, we considered a peak insulin level > 100 uIU/mL to be ‘pathological’. This cut-off is among those validated in the medical literature, and is not arbitrarily determined for the classification of OGTT results.

### 2.2. Cardiovascular Risk Indices

The cardiovascular risk indices calculated in our study were as follows:-Homeostasis model assessment of insulin resistance index (HOMA-IR) [10], calculated with the following formula: (Fasting plasma insulin × fasting plasma glucose)/22.5. This index combines basal blood glucose and insulin values to provide an indirect estimate of hyperinsulinism and increased insulin resistance. In general terms, HOMA-IR higher than 2.5 is considered pathological, although the cut-offs used may vary based on age, ethnicity, and gender.-Triglyceride glucose index (TyG) [11,12], calculated with the following formula: Ln [TG (mg/dL) × FPG (mg/dL)]/2; this index combines the value of triglycerides with glucose levels, incorporating two of the diagnostic criteria for the metabolic syndrome; it may be associated with increased cardiovascular risk when >4.5 [13].-Triglyceride to HDL ratio [14], a useful marker of cardiovascular risk applied also in overweight children; this index could be considered increased when >2 [15].-TyG-BMI [16], calculated with the following formula: TyG Index × BMI (kg/m^2^); this derives from the TyG index in association with patient’s BMI; however, there are no specific cut-offs available for its interpretation.-Visceral adiposity index (VAI) [17], calculated as reported in Figure 1; it describes how all determinants of the metabolic syndrome increase their absolute value in the presence of a higher VAI (>2.5) [18].

-Lipid accumulation product index (LAP) [19], calculated with the following formula: (LAP = (WC (cm) − 65) × TG (mmol/L)) for males, and (LAP = (WC (cm) − 58) × TG (mmol/L)) for females; we usually consider ‘pathological’ an LAP > 30; in adults it could be considered ‘pathological’ if higher than 56.7 for men and higher than 30.4 for women [20].-Waist/hip ratio (WHR) [21], calculated with the following formula: waist circumference (cm)/hip circumference (cm); its increase is associated with higher cardiovascular risk in both men and women [22], and is generally considered ‘pathological’ when exceeding the 95th percentile for age and sex, as there is no unique and universally accepted cut-off value for all clinical contexts.-Waist/height ratio (WHtR) [23], calculated with the following formula: waist circumference (cm)/height (cm); usually, a value > 0.5 is associated with obesity [24].-Fatty liver index (FLI) [25], an algorithm used to estimate the presence of fatty liver disease and hepatic steatosis: it is calculated using the formula reported in Figure 2, that incorporates several parameters such as BMI, waist circumference, triglycerides, and gamma-glutamyl transferase (GGT) levels. In pediatric patients, an FLI exceeding 30 is considered pathological, as it is associated with hepatic steatosis and increased cardiovascular risk.

-Hepatic steatosis index (HSI) [26], calculated using a formula based on BMI, waist circumference, and serum levels of AST (aspartate aminotransferase) and ALT (alanine aminotransferase), as reported in Figure 3. Similar to the FLI, an HSI > 30 is considered associated with hepatic steatosis and increased cardiovascular risk.

-Alanine aminotransferase/aspartate aminotransferase (AST/ALT) ratio, primarily studied in rheumatologic diseases to predict the risk of poor response to pharmacological treatments and risk of coronary artery damage [27,28]; in this context, a ratio above 1 is considered elevated.-Atherogenic index of plasma (AIP) [29,30], used to assess the risk of cardiovascular diseases, based on lipid levels in the blood and calculated as follows: [AIP = log10 (triglyceride/HDL cholesterol)]. In our study, we considered a cut-off of 0.1 to identify patients with increased cardiovascular risk.-In addition to the aforementioned indices, we also assessed, for each patient, the association between hyperinsulinemia and presence of elevated blood pressure (above the 95th percentile) [31] and hepatic steatosis, evaluated by abdominal ultrasound.

### 2.3. Statistical Analysis

The statistical analysis was performed using IBM SPSS software (version 20). Continuous normally distributed variables were described as mean values ± standard deviation (SD), the other continuous data as median ± interquartile range (IQR). To describe dichotomous data, we used numerosity and percentages. Groups of continuous variables were compared with Student *t*-test (normally distributed data) and Mann–Whitney U Test (not normally distributed data). Dichotomous variables were compared with Chi Square test or Fisher exact test (if numerosity of one of the compared groups was less than 6). We performed a linear logistic regression with continuous variables that reached a *p* value less than 0.2 at the univariate analysis. We calculated an ROC curve for those variables that resulted independently associated with hyperinsulinemia at the linear regression and calculated the AUC to determine their accuracy in predicting hyperinsulinism (insulin peak above 100 uIU/mL). ROC curves were also used to choose the best cut-off values for all variables using the Youden index. We used the calculated cut-off to transform the continuous dichotomous variables. Finally, we performed a multiple logistic regression adjusted for sex and age to identify the variables independently associated with hyperinsulinism. A *p* < 0.05 was considered statistically significant.

### 2.4. Ethical Approval

Ethics committee approval was not obtained because the General Authorization to Process Personal Data for Scientific Research Purposes (Authorization No. 9/2014) states that retrospective archival studies using ID codes that prevent direct tracing of data to the subject do not require a formal ethics approval. However, all parents of recruited patients were informed about the purpose of this study and signed an informed written consent for authorizing the access to children’s medical records and processing their personal data. Of all patients considered, none refused to participate to our study.

## 3. Results

We enrolled 139 children (58 males, 81 females) with a mean age of 12.1 ± 2.9 years and a mean body weight of 65.21 ± 19.3 kg. Out of the 139 children analyzed, 95 (68.35%) showed a peak insulin level > 100 uIU/mL during the OGTT, while in 44 cases (31.65%) the suspected hyperinsulinemia was not confirmed. All children included in this study exhibited normal renal function, assessed through measurements of serum creatinine and calculation of the estimated glomerular filtration rate. The general characteristics of our cohort are summarized in Table 1. 

All our patients were considered to have acquired hyperinsulinism, related to unhealthy dietary habits or unhealthy lifestyles. In none of the cases it was congenital, as signs and/or symptoms of neonatal hypoglycemia and other suggestive criteria for inherited conditions were absent [29]; therefore, genetic testing assays were not performed.

Patients with pathological OGTT were considered to be at increased cardiovascular risk; therefore, we calculated the various indices, previously described, assessing differences between patients with normal insulinemic response and those with insulinemia > 100 uIU/mL.

Comparing the two groups, we found statistically significant differences in weight, BMI, WC, HC, IGF-1, uric acid, VLDL, triglycerides, and basal insulin levels; *p* values are reported in Table 1.

After analyzing auxological and laboratory data, we calculated the previously described cardiovascular risk indices for each patient to assess differences between confirmed hyperinsulinemic and non-hyperinsulinemic patients. Comparing the two groups, we found statistically significant differences in HOMA-IR, TyG, TyG-BMI, VAI, LAP, FLI, HIS, and Triglycerides/HDL, while WHtR and WHR did not differ significantly. All *p* values are reported in Table 2.

At the linear logistic regression, we found that IGF-1, HOMA-IR, and ALT/AST ratio were independently associated with a confirmed hyperinsulinism (Table 3).

For all continuous variables correlated with hyperinsulinism, we created the ROC curves with the aim of finding the best cut-off points to differentiate patients with hyperinsulinemia from those without, and calculate the diagnostic accuracy using AUCs (Figure 4). From our analysis, HOMA-IR emerged as the best parameter for diagnosing hyperinsulinism (Table 4).

Thereafter, we calculated the optimal cut-off values of those parameters that were independently associated with hyperinsulinism at the linear logistic regression (Table 5).

Using the identified cut-off, we transformed the variables from continuous to dichotomous, and performed a multinomial logistic regression, adjusting our data for sex and age. IGF-1 levels higher than 203 ng/mL and HOMA-IR values higher than 6.2 were respectively associated with a 9-times and 18-times higher risk of confirmed hyperinsulinism, compared to those with IGF levels less than 203 and HOMA-IR less than 6.2. The other investigated parameters were not significantly related to hyperinsulinism, and based on the results of our statistical analysis, they could not predict the presence of hyperinsulinemia in our cohort.

All analyzed children underwent a comprehensive endocrinological follow-up and are currently being monitored at our center; during the follow-up period, none of the patients experienced major cardiovascular events.

Given the results obtained and considering that it was not possible to match patients by sex and age due to sample size, we conducted subgroup analyses to assess whether differences in sex and age (prepubertal vs. pubertal patients, using 9 years as the age limit) could act as confounding factors for hyperinsulinemia.

When comparing sex in our populations in the univariate analysis, we observed no differences regarding the mean age. Furthermore, it appears that a smaller number of male subjects had IGF-1 values > 203: 27/58 (47%) vs. 60/81 (74%); *p* < 0.0001. A similar but non-significant trend was also observed for LAP > 15.9, although it did not reach a statistical significance (*p* = 0.08). No significant difference was found between sexes for the prevalence of hyperinsulinemia (*p* = 0.17).

Conducting a multivariate analysis including IGF-1 and LAP and correcting for age, we confirmed that IGF-1 values > 203 were more strongly associated with females, while LAP was not significantly associated with patients’ sex.

In the univariate analysis, it was found that patients over 9 years were more likely to have BMI < 25.72, WC > 82.5, IGF-1 > 203, HOMA-IR > 2.62, TyG-BMI > 131, LAP > 0.02, and WHR > 0.03. We also observed a trend, although not statistically significant, towards a higher likelihood of hyperinsulinemia (*p* = 0.06). We also performed a multivariate analysis correcting for sex and including all factors that were associated with age > 9 years in the univariate analysis with a significative *p* value: at the multivariate analysis, we confirmed that BMI, IGF-1, and WHR were the only factors associated with age > 9 years after correction for confounding factors.

Finally, we conducted further subgroup analyses to evaluate which indices were correlated with insulin resistance in females, in males, and in those aged more or less than 9 years. In female patients, at the multivariate analysis, correcting for age and including factors with a significative *p* value at the univariate analysis, we found that only HOMA-IR was associated with hyperinsulinemia. We repeated the same analysis in male patients, finding that only IGF-1 was associated with insulin resistance. In patients under 9 years, only HOMA-IR was statistically associated with insulin resistance, while in the subgroup over 9 years, only IGF-1 and HOMA-IR were effectively associated with hyperinsulinemia.

## 4. Discussion

The incidence of obesity in childhood is currently increasing, potentially leading to overt psychophysical issues during school age and several medical diseases in early adulthood [30,31,32,33,34]. In Italy, approximately 21.3% and 9.3% of school-aged children are classified as overweight and obese, respectively [34]. The Position Paper of the European Childhood Obesity Group and the European Academy of Pediatrics state that the main cause of overweight in pediatric age is the lack of a regular physical activity. Therefore, public health interventions are needed since school age to increase the level of physical activity and reduce a potential insulin resistance with hyperinsulinemia [35,36].

The increasing rate of obesity and the ever-growing healthcare expenditure to treat overweight-related complications make early diagnosis and appropriate prevention strategies imperative even during childhood. A recent consensus position statement of the Italian Society of Pediatric Endocrinology and diabetology, Italian Society of Pediatrics, and Italian Society of Pediatric Surgery has highlighted that a healthy lifestyle, a balanced diet, and a regular physical activity are milestones for managing pediatric patients with obesity, while pharmacological therapies should be introduced as second-line tools, and bariatric surgery reserved for those adolescents with severe obesity resistant to all other previous treatments [37]. Childhood obesity contributes to the development of early onset-cardiovascular diseases, though children can develop heart diseases due to other severe health issues, including Kawasaki disease or acute rheumatic fever, and if affected by peculiar lysosomal storage, disorders involving the heart [37,38,39,40,41,42]. The specific interaction of dietary macronutrients and the endocrine system might have a role in the etiology of obesity, and different genetic disorders display subverted processing of lipids following mutations in several genes involved in autoinflammation, as for children with mevalonate kinase deficiency [41]. Indeed, childhood obesity is related to a powerful array of cardiovascular risk factors, with increased triglycerides and increased lipoproteins being among the most relevant players, even if hypertriglyceridemia can be also a marker of systemic fatal complications in many pediatric diseases [40]. Hyperinsulinemia is an independent cardiovascular risk factor; therefore, identifying indices that can predict it in the pediatric population is crucial for the development of effective preventive strategies [1,3]; physical exercise training, particularly aerobic, should be capable of normalizing both auxological parameters related to overweight and obesity as well as improving insulin resistance in overweight or obese children and adolescents, though dietary interventions remain crucial [37]. 

During pediatric age, hyperinsulinemia can have a genetic basis, being caused by molecular alterations in insulin secretion or in insulin receptor response; these alterations can be classified as isolated hyperinsulinemia or as part of the clinical presentation of various genetic syndromes [43]. However, congenital hyperinsulinemia is a rare condition, and most of the elevated insulin peaks detected during OGTT might be explained by improper dietary habits and unhealthy lifestyles [8,34]; the bidirectional relationship between obesity and hyperinsulinemia is well established in the scientific literature. Obesity contributes to the development of insulin resistance, leading to compensatory hyperinsulinemia as the body attempts to maintain normal blood glucose levels [44]. Conversely, hyperinsulinemia promotes weight gain by stimulating lipogenesis and inhibiting lipolysis, thereby exacerbating the obesity phenotype [45]. This reciprocal interaction forms a vicious cycle, perpetuating both conditions. Different studies have elucidated this complex interplay, highlighting the pivotal role of hyperinsulinemia in the pathogenesis of obesity and vice versa [43,44,45,46].

The exact significance of hyperinsulinemia in pediatric patients is controversial, although it is a well-validated cardiovascular risk factor: in fact, several studies have highlighted that increased insulin secretion is not necessarily associated with weight gain or worsening BMI over time [47]. Moreover, elevated insulin levels have been shown to correlate with decreased cardiac diastolic function in the complete absence of any clinical symptoms [48]. 

In our study, we have applied various cardiovascular risk indices commonly used in adulthood using patients’ auxological parameters and their blood tests to assess their diagnostic accuracy in the pediatric age: our results have shown that these indices, with the exception of HOMA-IR, are not significantly associated with hyperinsulinemia in children and adolescents, and that they could not predict the risk of developing a metabolic syndrome or further cardiovascular comorbidities in childhood. 

HOMA-IR has been proven to be an excellent marker of hyperinsulinemia in the pediatric age: the higher predictive value of HOMA-IR is associated with a close relationship between basal and stimulated insulin level after OGTT [49], as clearly shown by several studies [11]. Our statistical analysis has revealed that HOMA-IR is the only index that confirms its diagnostic accuracy in diagnosing hyperinsulinemia for both children and adolescents. Therefore, it could be applied to estimate the cardiovascular risk in this category of patients. 

A recent study [50] analyzed 3203 Chinese children aged 6 to 18 years, determining the best predictive cut-offs for metabolic syndrome: the authors highlighted that the optimal HOMA-IR cut-off for diagnosis was 2.3 in the total participants, 1.7 in prepubertal children, and 2.6 in pubertal adolescents (>9 years); moreover, 44.3% of obese patients had values > 3. Our statistical analysis confirmed these results: indeed, in our population, the optimal cut-off was found to be 2.62, while the multivariate analysis showed that HOMA-IR > 6.2 was associated with risk of hyperinsulinemia approximately 18 times higher. 

In addition, the importance of HOMA-IR lies in its ability to provide a quick and immediate estimate of insulin secretion and insulin resistance without resorting to investigative procedures like the OGTT [49], which are not always easily performed in the pediatric age [8]. This explains why this index is one of the most commonly used in clinical practice, widely employed even in the pediatric population.

Early diagnosis of hyperinsulinemia is crucial to enable early treatment with lifestyle modifications and improved dietary balance [35,36]. Indeed, while increased insulin levels are a well-established cardiovascular risk factor in adults, an elevation in insulin levels during pediatric age leads to worsened long-term outcomes. A recent study (The Pune Children’s Study) [51] examined the influence of glycemia, insulin, and HOMA-IR in a cohort of 8-year-old children, re-evaluated at the age of 21. The authors highlighted that values observed during pediatric age were statistically correlated with those measured in young adults. Furthermore, the authors found that higher levels of HOMA-IR were associated with a worse cardiovascular risk profile (assessed through measurement of blood pressure, plasma lipids, carotid intima-media thickness, and arterial pulse wave velocity). In fact, the authors showed that prepubertal glucose and insulin metabolism were associated with abnormal markers of atherosclerosis and early cardiovascular risk.

Less information is available about the application of other cardiovascular risk indices in the pediatric age, because they were only validated in the adult population and mostly applied for research purposes.

Regarding TyG, an analysis conducted on 367 children and adolescents showed that the cut-off of 7.96 had the best sensitivity and specificity (65% and 58%, respectively) for diagnosing insulin resistance, confirmed by HOMA-IR [52]. This cut-off was much higher than the one of 4.5 used for adults, which had higher rates of concordance. Our analysis did not confirm the triglyceride/HDL ratio as an index of hyperinsulinemia and insulin resistance. However, the study by de Giorgis et al. [14] highlighted that an alteration in this parameter could lead to changes in carotid intima-media thickness. Increasing our sample size may yield different results. Despite these considerations, the triglyceride/HDL ratio is a well-validated cardiovascular risk index [15] and should be assessed in patients at risk of developing metabolic syndrome, such as children with hyperinsulinemia [1,4]. VAI is a further parameter related to auxological and laboratory parameters, which can be considered an indirect index of dysfunction of the endocrine adipose tissue, identifying a higher risk of developing diabetes [53]. Amato et al. conducted a study on 1764 adult patients, calculating VAI and highlighting that a cut-off of 2.52 in subjects under 30 years had a sensitivity of 100% and specificity of 99.45% to identify adipose tissue dysfunction and cardiometabolic risk, while the optimal cut-off decreased with increasing age [54]. Vizzuso et al. have, however, demonstrated that the optimal cut-off for VAI in obese pediatric patients (with a mean age of 11 years) is 1.775, with differences between males (lower indices) and females [55]. Our statistical analysis has highlighted that the best cut-off for VAI was 1.66, confirming previous authors’ data. 

LAP is another index that combines auxological parameters and laboratory data, estimating the accumulation of adipose tissue in the body, showing an excellent prediction of non-alcoholic fatty liver disease (NAFLD) in adults [56]. In childhood, there is limited evidence regarding this correlation, but the only published pediatric study analyzed 80 patients diagnosed with obesity, undergoing liver ultrasound screening for NAFLD; the authors demonstrated that LAP > 42.7 had a sensitivity of 53.7% and specificity of 84.6%, respectively, for diagnosing NAFLD [57]. Although our statistical analysis did not consider the correlation between LAP and steatosis, this index was not associated with hyperinsulinemia in our cohort; the best cut-off resulting from our multivariate analysis was 15.9, which should not allow a diagnosis of hepatic steatosis. In our cohort, both WHtR and WHR were also found inadequate to differentiate patients with or without hyperinsulinemia. In our multivariate analysis, WHR was not associated with insulin resistance. However, it is possible that our sample (139 children with suspected hyperinsulinemia) may not be sufficiently large for a proper validation of this index, as studies on numerous pediatric patients with comorbidities have shown that WHR is excellent to assess the cardiovascular risk if children have comorbidities [58], predicting the risk of developing a metabolic syndrome [24]. 

FLI and HSI are two commonly used clinical indices associated with metabolic syndrome and hepatic steatosis [25,26]. In a study conducted on 95 children aged between 5 and 15 years, all with BMI above the normal upper limits (>85th percentile corrected for sex and age), a high concordance between FLI and hepatic steatosis was found and confirmed by ultrasound in 36 children, with an AUROC of the FLI for predicting NAFLD of 0.692 (95% CI: 0.565–0.786); the cut-off of 30, analogous to that used in our study, showed sensitivity values of 58.3% and specificity of 69.4% [25]. Our multivariate analysis did not confirm the diagnostic accuracy of these two indices in assessing the risk of hyperinsulinemia; however, the correlation with hepatic steatosis is widely demonstrated in the medical literature. Therefore, although they are not associated with insulin dysregulation, both could be used clinically to study the cardiovascular risk in pediatric patients.

The measurement of transaminases is also included in the most commonly used screening hematochemical tests in clinical practice. The AST/ALT ratio has been proven to be an effective index in predicting the risk of coronary artery disease in patients with Kawasaki disease [27]. Additionally, studies conducted on adult patients have shown a higher correlation between this index and a diagnosis of hyperinsulinemia [28]. In our cohort of pediatric patients with hyperinsulinemia, we did not confirm this finding. This could be because hyperinsulinemia is not necessarily associated with hepatic steatosis [6], and it is possible that an increase in transaminases is only evident in more severe forms. 

AIP is another marker currently undergoing validation in the pediatric population. Dag et al. [30] analyzed 136 adolescents (83 obese and 53 healthy controls) aged between 10 and 17 years, demonstrating that this index is higher in patients with hepatic steatosis and increased BMI compared to healthy controls. The same authors highlighted a significant statistical relationship between AIP and basal insulin, identifying this index as a potential marker for hyperinsulinemia. Our statistical analysis did not confirm this result, despite the sample size of our study; it is possible that by increasing the sample size, different results may be obtained, and future analyses are necessary to confirm the clinical relevance of this index.

The significant association between IGF-1 and hyperinsulinemia is already known, as they are closely related in intracellular signaling pathways: normal levels of IGF-1 contribute to the normal development and function of the cardiovascular system [59]. In our sample, an IGF-1 level higher than 203 was found to be associated with a 9-fold increased risk of hyperinsulinemia. Nevertheless, IGF-1 is not frequently measured in obese patients, and its use is largely referred to linear growth assessment and evaluation of growth hormone activity [60]. Despite this, a recent systematic review has analyzed the relationship among IGF-1, growth hormone, and obesity, highlighting that a modulation of these hormones could prevent the progression of metabolic syndrome and associated cardiovascular complications [61]. The development of pediatric cardiovascular risk indices might take into consideration this parameter combined with other anthropometric measurements.

Of course, our study has some limitations. Firstly, it is a retrospective single-center study, although our conclusions lay the groundwork for larger-scale prospective studies. The sample size is another limitation, so it is possible that the results of our statistical analysis may not be applicable to the general population. Despite these considerations, our study also has some strengths. We have contributed to defining possible cut-off values for common cardiovascular risk indices in a cohort of children and adolescents under 18 years. Currently, there are no well-defined normal ranges for these indices in the pediatric population. Furthermore, the OGTT is the gold standard for assessing hyperinsulinemia, not only in pediatric patients but also in adults. The implementation of specific cardiovascular risk indices for the pediatric population may, in the future, estimate the prevalence of hyperinsulinemia under 18 years, rendering unnecessary the execution of OGTT. 

## 5. Conclusions

In our pediatric cohort of patients with suspected hyperinsulinism, the commonly used indirect indices of cardiovascular risk in the adult population, excluding HOMA-IR, were not accurate in confirming hyperinsulinemia. Comparing patients with and without hyperinsulinemia, we found statistically significant differences in weight, BMI, WC, HC, IGF-1, uric acid, VLDL, triglycerides, and basal insulin levels. The multivariate analysis highlighted that an IGF-1 level higher than 203 ng/mL and HOMA index higher than 6.2 are respectively associated with a 9-times and 18-times higher odds ratio for hyperinsulinism. 

The results of our statistical analysis represent the starting point for future prospective studies to analyze the cardiovascular risk in pediatric patients displaying hyperinsulinemia and establish the optimal cut-offs of such indices.

## Figures and Tables

**Figure 1 diseases-12-00119-f001:**
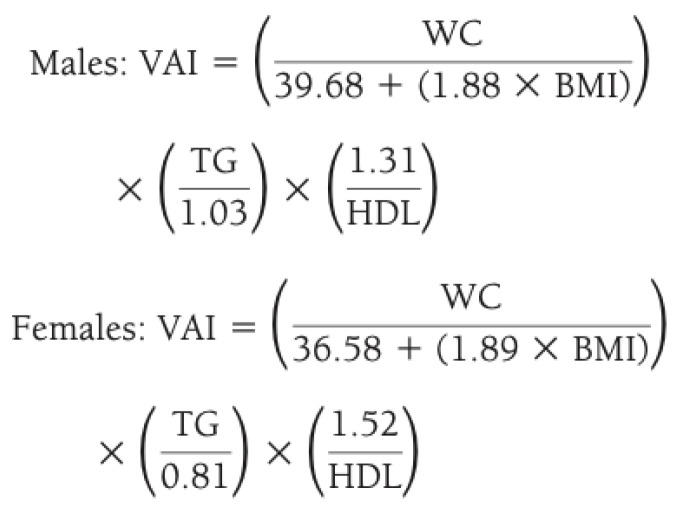
Method for calculating visceral adiposity index (VAI) by combining anthropometric and blood exams. Legend: WC waist circumference; BMI body mass index; TG triglycerides; HDL high density lipoprotein.

**Figure 2 diseases-12-00119-f002:**
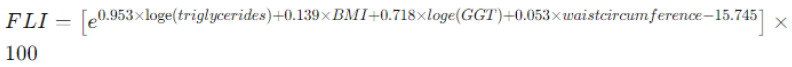
Method for calculating fatty liver index (FLI) by combining anthropometric and blood exams. Legend: BMI body mass index, GGT gamma-glutamyl transpeptidase.

**Figure 3 diseases-12-00119-f003:**
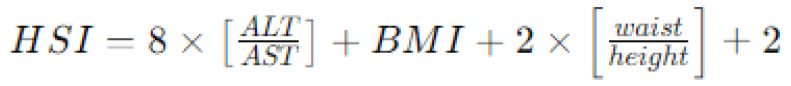
Method for calculating hepatic steatosis index (HSI) by combining anthropometric and blood exams. Legend: ALT aspartate aminotransferase, ALT alanine aminotransferase; BMI body mass index.

**Figure 4 diseases-12-00119-f004:**
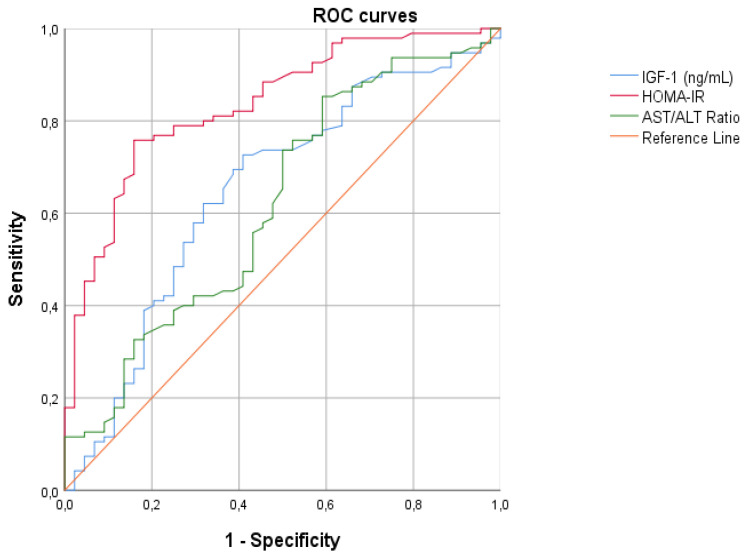
ROC curves assessing the cardiovascular risk and laboratory parameters of patients. Legend: IGF-1 insulin growth factor-1; HOMA-IR homeostasis model assessment of insulin resistance index; AST/ALT ratio alanine aminotransferase/aspartate aminotransferase ratio.

**Table 1 diseases-12-00119-t001:** Anthropometric and laboratory characteristics of patients with suspected hyperinsulinemia, confirmed or not confirmed following the OGTT.

Variable	Population	Insulin Peak > 100 uIU/mL	Normal Insulin Peak	*p* Value
Age (years) ± SD	12.1 ± 2.9	12.1 ± 2.9	12.1 ± 2.9	0.52
Sex (male) N (%)	58 (41.7)	36 (37.9)	22 (50)	0.17
Weight (kg) ± SD	65.21 ± 19.3	68.02 ± 19.7	59.16 ± 17.2	0.01
Height (cm) ± SD	150.74 ± 13.9	151.89 ± 12.7	148.26 ± 16.1	0.19
BMI (kg/cm^2^) ± SD	28.06 ± 4.3	28.85 ± 4.5	26.36 ± 3.3	<0.001
WC (cm) ± SD	86.66 ± 12.3	88.20 ± 12.3	83.33 ± 11.6	0.03
WC > 75° N (%)	130 (93.5)	90 (94.7)	40 (90.9)	0.46
WC > 90° N (%)	94 (67.6)	66 (69.5)	28 (63.6)	0.49
HC (cm) ± SD	97.96 ± 11.3	99.60 ± 11.1	94.41 ± 11.1	0.01
DM familiarity N (%)	112 (80.6%)	75 (78.9)	37 (84.9)	0.47
GOT (IU/L) ± SD	23.1 ± 6.3	22.4 ± 5.0	24.59 ± 6.8	0.07
GPT (IU/L) ± SD	21.66 ± 12.4	22.28 ± 13.8	20.34 ± 8.8	0.32
Uric acid (mg/dL) ± SD	4.87 ± 1.2	5 ± 1.3	4.58 ± 1.0	0.05
TSH (uIU/mL) ± SD	2.97 ± 1.7	3.09 ± 1.8	2.70 ± 1.4	0.16
fT3 (pg/mL) ± SD	4.10 ± 0.5	4.12 ± 0.5	4.06 ± 0.5	0.47
fT4 (pg/mL ± SD	11.31 ± 1.6	11.19 ± 1.4	11.56 ± 1.9	0.27
IGF-1 (ng/mL) ± SD	264.4 ± 117.4	280.87 ± 115.4	228.86 ± 115.1	0.02
HbA1C (mmol/mol) ± SD	35.95 ± 3.3	36.14 ± 3.4	35.55 ± 3.1	0.32
Total cholesterol (mg/dL) ± SD	154.78 ± 27.5	155.80 ± 29.7	152.59 ± 22.2	0.48
HDL (mg/dL) ± SD	45.93 ± 9.4	45.45 ± 9.7	46.95 ± 8.8	0.37
LDL (mg/dL) ± SD	91.32 ± 21.8	92.28 ± 22.9	89.27 ± 19.5	0.43
VLDL (mg/dL) ± SD	19.65 ± 15.3	21.19 ± 17.2	16.26 ± 8.9	0.03
Triglycerides (mg/dL) ± SD	98.08 ± 76.1	105.41 ± 86.2	82.25 ± 44.0	0.04
Glucose (mg/dL) ± SD	85.42 ± 7.1	86.05 ± 7.2	84.05 ± 6.9	0.12
Insulin (uUI/m) ± SD	20.96 ± 21.38	25.48 ± 24.4	11.19 ± 5.1	<0.001
GGT (u/L) ± SD	17.78 ± 10.61	17.78 ± 9.51	17.80 ± 12.79	0.994
Blood pressure > 95th N (%)	22	17 (17.89)	5 (11.36)	0.45
Steatosis N (%)	90 (64.75)	84 (88.42)	6 (13.64)	0

Legend: OGTT oral glucose tolerance test; BMI body mass index; WC waist circumference; HC hip circumference; DM diabetes mellitus; GOT serum glutamic oxaloacetic transaminase; GPT serum glutamate-pyruvate transaminase; TSH thyroid-stimulating hormone; fT3 triiodothyronine; fT4 thyroxine; IGF-1 insulin growth factor-1; HbA1C glycated hemoglobin; LDL low density lipoprotein; HDL high density lipoprotein; VLDL very low density lipoprotein; GGT gamma-glutamyl transferase.

**Table 2 diseases-12-00119-t002:** Comparison of the main cardiovascular risk indices between the groups of patients with suspected hyperinsulinemia, confirmed or not, following the OGTT.

CV Index	Population Mean ± SD	Insulin Peak > 100 Mean ± SD	Normal Insulin Peak Mean ± SD	*p* Value
HOMA-IR	3.76 ± 3.99	4.56 ± 4.56	2.04 ± 1.08	<0.001
TyG	4.44 ± 0.3	4.48 ± 0.3	4.36 ± 0.2	0.01
TyG-BMI	124.67 ± 20.9	129.1 ± 21.1	115.1 ± 16.9	<0.001
Triglycerides/HDL	2.19 ± 1.95	2.53 ± 2.49	1.86 ± 1.20	0.034
VAI	1.48 ± 1.0	1.62 ± 1.1	1.19 ± 0,	0.01
LAP	29.1 ± 28.4	32.64 ± 31.2	21.36 ± 19.2	0.01
WHtR	0.58 ± 0.1	0.58 ± 0.1	0.56 ± 0.1	0.18
WHR	0.65 ± 0.1	0.66 ± 0.1	0.64 ± 0.1	0.11
FLI	3.72 ± 6.74	4.57 ± 7.78	1.89 ± 2.93	0.004
HSI	29.61 ± 6.89	30.98 ± 6.84	26.65 ± 6.07	0.0003
AST/ALT ratio	0.92 ± 0.32	0.96 ± 0.34	0.82 ± 0.26	0.01
AIP	0.27 ± 0.26	0.31 ± 0.27	0.20 ± 0.24	0.02

Legend: HOMA-IR homeostasis model assessment of insulin resistance index; TyG triglyceride glucose index; BMI body mass index; VAI visceral adiposity index; LAP lipid accumulation product index; WHR waist/hip ratio, WHtR waist/ height ratio; FLI fatty liver index; HIS hepatic steatosis index; AST/ALT ratio alanine aminotransferase/aspartate aminotransferase ratio; AIP atherogenic index of plasma.

**Table 3 diseases-12-00119-t003:** Parameters associated with hyperinsulinemia based on the logistic linear regression analysis.

Parameter	*p* Value	Parameter	*p* Value
BMI (kg/cm^2^)	0.962	HOMA-IR	0.021
WC (cm)	0.948	TyG	0.768
HC (cm)	0.946	TyG-BMI	0.882
TSH (IU/mL)	0.711	VAI	0.288
IGF-1 (ng/mL)	0.019	LAP	0.312
Uric acid (mg/dL)	0.904	WHtR	0.764
VLDL (mg/dL)	0.100	WHR	0.985
Triglycerides (mg/dL)	0.175	FLI	0.924
HSI	0.312	AST/ALT	0.049
AIP	0.206	HDL/Triglycerides	0.442

Legend: BMI body mass index; WC waist circumference; HC hip circumference, TSH thyroid stimulating hormone; IGF-1 insulin growth factor-1; VLDL very low density lipoprotein; HOMA-IR homeostasis model assessment of insulin resistance index; TyG triglyceride glucose index; BMI body mass index; VAI visceral adiposity index; LAP lipid accumulation product index; WHR waist/hip ratio; WHtR waist/height ratio; FLI fatty liver index; HIS hepatic steatosis index; AST/ALT ratio alanine aminotransferase/aspartate aminotransferase ratio; AIP atherogenic index of plasma.

**Table 4 diseases-12-00119-t004:** Diagnostic accuracy (measured as AUC—area under the curve) for indirect indices and laboratory parameters of cardiovascular risk that resulted associated with hyperinsulinism at the linear logistic regression.

Parameter	AUC	95% Confidence Interval	*p*
IGF-1 (ng/mL)	0.650	0.549–0.751	0.005
HOMA-IR	0.836	0.767–0.906	>0.0001
AST/ALT ratio	0.620	0.517–0.722	0.023

Legend: AUC area under the curve; IGF-1 insulin growth factor-1; HOMA-IR homeostasis model assessment of insulin resistance index; AST/ALT ratio alanine aminotransferase/aspartate aminotransferase ratio.

**Table 5 diseases-12-00119-t005:** Best cut-off values evaluated by means of the Youden index and multiple logistic regression analysis adjusted for age and sex to evaluate the association of laboratory parameters and indices of cardiovascular risk in our cohort of children with hyperinsulinism.

Variable	Best Cut-Off	*p*	OR (95%IC)
Age		0.33	1.090 (0.917–1.296)
Male sex		0.63	1.270 (0.472–3.420)
IGF-1 (ng/mL)	202	0.005	5.359 (1.674–17.153)
HOMA-IR	2.62	<0.0001	15.774 (5.537–44.936)
ALT/AST ratio	0.69	0.40	1.616 (0.529–4.934)

Legend: IGF-1 insulin growth factor-1; HOMA-IR homeostasis model assessment of insulin resistance index; AST/ALT ratio alanine aminotransferase/aspartate aminotransferase ratio.

## Data Availability

No new data were created.

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
