# Peer review of "Cardiometabolic Risk Assessment in a Cohort of Children and Adolescents Diagnosed with Hyperinsulinemia"

_diseases, 2024, doi:10.3390/diseases12060119_

Round 1
Reviewer 1 Report (Previous Reviewer 1)
Comments and Suggestions for Authors
The revised version signficantly improved and merit publication
Author Response
Dear Reviewer,
Thank you for your comments, which have helped us improve our manuscript. We are pleased that the revised version has met your expectations.
Kind regards
Reviewer 2 Report (New Reviewer)
Comments and Suggestions for Authors
The manuscript is very interesting because it deals with a progressively increasing problem at an Italian and global level, particularly in developed and developing countries, i.e. infantile hyperinsulinemia resulting from an incorrect lifestyle (excessive intake of calories, particularly in the form of carbohydrates, and insufficient physical activity). On the other hand, there is extensive scientific literature that supports hyperinsulinemia associated with insulin resistance as a risk factor for cardiovascular disease, type 2 diabetes, cellular senescence, cognitive deficits and dementia. Given that the authors have the values of insulinemia, which is high in the majority of cases studied, it would be useful for them to stress in the discussion the topic of hyperinsulinemia as an independent risk for cardiovascular diseases and more. Furthermore, it has been shown that the ratio of triglycerides to HDLc is predictive of cardiovascular events in the adult population.
From the results reported in the table, we note that the value of this ratio obtained from the average values of triglycerides and HDLc is 2.32 in subjects with increased insulinemia and 1.75 in subjects with normal insulinemia, a difference which could be statistically significant. For this reason, I believe that the authors should add this parameter in the data analysis.
Comments on the Quality of English Language
The English is sufficiently good.
Author Response
Dear Reviewer,
Thank you for your insightful comments, which have greatly contributed to the enhancement of our manuscript.
We have incorporated the triglycerides/HDL ratio into our statistical analysis; however, it did not emerge as statistically significant in the linear logistic regression. Nonetheless, in the discussion section, we have integrated information regarding this well-validated index of cardiovascular risk in the adult and pediatric population.
We sincerely appreciate your invaluable assistance in finalizing our manuscript and hope for the opportunity to collaborate again in the future.
Kind regards
Dr Giorgio Sodero
This manuscript is a resubmission of an earlier submission. The following is a list of the peer review reports and author responses from that submission.
Round 1
Reviewer 1 Report
Comments and Suggestions for Authors
The study addressed an important issue about children insulin resistance and metabolic syndrome. The study number is small. However, the conclusion was solid and might initiate large scale study.
The study is well done. I only have one question
Does the study individual have diagnosis of T1DM or Type 2 DM or even have Major adverce cerebral cardiovascular event in the following period?
Author Response
Reviewer 1
The study addressed an important issue about children insulin resistance and metabolic syndrome. The study number is small. However, the conclusion was solid and might initiate large scale study.
The study is well done. I only have one question
Does the study individual have diagnosis of T1DM or Type 2 DM or even have Major adverce cerebral cardiovascular event in the following period?
Dear Reviewer,
We appreciate your comments on our manuscript. While our sample size is limited, our results lay the groundwork for future prospective studies examining cardiovascular risk indices in pediatric populations.
Regarding the classification of our patients, they are children and adolescents with isolated hyperinsulinism; none of our patients received a diagnosis of diabetes, yet all are at subsequent risk of developing metabolic syndrome and type II diabetes. The children are currently under follow-up at our center, and none have experienced major cardiovascular events thus far.
Thank you once again for your time, and we hope that the revised version of our manuscript meets your expectations.
Kind regards
Reviewer 2 Report
Comments and Suggestions for Authors
There are some difficulties with English presentations (singulars and plurals confused etc.) but this can be addressed in the revised manuscript
Author Response
Reviewer 2
Dear Reviewer,
Thank you for your comments, which have helped us improve the quality of our manuscript. Below, you will find a point-by-point analysis of your comments along with our responses.
While this manuscript has potential to inform new measures for hyperinsulinism this reviewer is confused regarding the focus of this manuscript. If it is congenital hyperinsulinism then the authors need to address why they did not perform the typical genetic tests (see references below) for this disorder and see how it correlated with their measures. Their measures alone would not address this question and the reason for omitting needs to be addressed.
Thank you for your observation. In fact, none of our patients had a diagnosis of congenital hyperinsulinemia; indeed, none of the children had a clinical history suggestive of congenital pathology (recurrent hypoglycemia, neonatal hypoglycemia, neurological signs and symptoms, alterations during perinatal period), and in all patients, hyperinsulinemia was considered acquired (and therefore related to dietary habits and lifestyle). Therefore, these are non-diabetic patients but with a significant risk of subsequently developing type II diabetes and metabolic syndrome, which are identified following OGTT execution (a test not easily reproducible in pediatric age due to children's cooperation, the frequent need for venous access, and serial blood draws); hence our need to apply common cardiovascular risk indices for adults to assess their correlation with hyperinsulinemia.
We understand your considerations about congenital hyperinsulinism; therefore, we have modified the article in several places, also integrating some of the articles you pointed out during the review, and specifying the lack of genetic testing in our patients.
Thank you for your valuable input, and we trust that these revisions address your concerns adequately.
Minor points:
Line 36: the term "hyperinsulinemia" is used is this meant to be "hyperinsulinism"?
In our manuscript, we have used the terms "hyperinsulinemia" and "hyperinsulinism" interchangeably. We have made several modifications to our article to ensure consistency. Thank you for bringing this to our attention.
The second main issue is why (if the authors are looking to PREDICT metabolic syndrome) the authors would include subjects that already appear to meet almost all the criteria for metabolic syndrome. The data should be reanalyzed with those subjects excluded.
According to data from the International Diabetes Federation, children aged 10 to 15 years meet the criteria for metabolic syndrome if they exhibit obesity with waist circumference > 90th percentile and two risk factors (blood pressure ≥130/85 mmHg, HDL ≤40 mg/dL, triglycerides ≥150 mg/dL, or fasting glucose ≥100 mg/dL). Notwithstanding, it is not possible to diagnose this syndrome in children under 10 years old due to the absence of age-specific reference values. Based on the criteria listed, none of our patients meets the defined diagnosis of metabolic syndrome; however, the presence of hyperinsulinemia detected during OGTT is considered a risk factor for the subsequent development of metabolic syndrome and type II diabetes. We have further clarified this information in the inclusion criteria of our study. Thank you for your observation.
Inclusion criteria include high BMI and WH ratio which are symptoms of metabolic syndrome so if that is what is being predicted these subjects should be excluded.
Rosenfeld E, Ganguly A, De Leon DD. Congenital hyperinsulinism disorders: Genetic and clinical characteristics. American Journal of Medical Genetics Part C: Seminars in Medical Genetics. 2019;181:682-92
As previously mentioned, an increase in BMI or waist circumference alone is not sufficient to diagnose metabolic syndrome in pediatric patients. Therefore, we did not exclude patients with an increased BMI, as they do not meet the criteria for metabolic syndrome. Additionally, we have included the article you mentioned during the review process in the discussion section. Thank you for your comment.
There are some difficulties with English presentations (singulars and plurals confused etc.) but this can be addressed in the revised manuscript
Thank you for your observations. We have revised the English in many parts of the manuscript. We appreciate your thorough review and look forward to hearing your thoughts on the revised manuscript.
Kind regards
Reviewer 3 Report
Comments and Suggestions for Authors
The paper is unclear in many passages and I have remarkable concerns about the numerosity of the sample and method using to assess hyperinsuliemia.
Major criticisms:
1) The scarce sample numerosity underpowers the analysis. did the authors conduct a sample size analysis to ensure statistical significancy of the sample and then of the analysis? Discuss this aspect eventually reporting similar analysis on the topic and referring to sample size of such studies. I would eventually state the scarce numerosity as a critical limitation of the study.
2) Could a different metabolic regulation occurring at puberty between males and females have impacted the results? Please address this aspect and eventually consider a further analysis separating males and females, to see if the variables considered are still correlated gender-wise.
3) Importantly, it's unclear how the authors assesses hyperinsulinemia (which should be the reference parameter to correlate with other variables); The authors state: "In the absence of a common guideline for interpreting insulin levels during OGTT we considered a peak insulin level >100 uIU/mL to be pathological, as commonly done in our center". Please: A) reference this passage (if possible) or further discuss, with citing appropriate literature, the rationale behind this methodological choice, which sounds arbitrary. B) Clearly state how you defined hyperinsulinemia in the patients considered for the analysis. For example you report in the inclusion criteria as well as in the discussion the phrase " suspected insulinism". Was the insulinism "suspected" or ascertained? Please be clear in the description of methodologies ant throughout the text. This aspect should be addressed and you should clearly state in the text if you are correlating clear hyperinssulinemima or suspected insulinemia The choice of parameters to define hyperinsulinemia is critical for correctness of the analysis and warrants reliability of results gained.
4) The authors are required to critically and clearly discuss the limitations of their work (some of them have been suggested in these comments).
5) State the type of study you are conducting (cross-sectional assessing correlations etc). For example, in the conclusion you state "Multivariate analysis highlighted that IGF-1 level higher than 203 ng/mL and HOMA index higher than 6.2 are respectively associated with a 9- and 18-times higher risk of hyperinsulinism". The association "marker-risk of something" can be assessed in perspective studies, in which patients are followed for a certain span o time. Pleasese address this aspect and clearly state your conclusion to convey the core message of your paper in order to make it useful for reseachers and clinicians.
6) The conclusions can be improved may be suggesting what you found positively associated with hyperinsulineima. Also, suggest the usefulness and relevancy of such analysis from a clinical standpoint and in the context of exsiting scenario (wherever appropriate, may be also in the discussion).
Minor criticisms
In the introduction and eventually in the discussion, the authors are asked to provide more information on the relationship obesity-hyperinsulinemia (what upregulates insulin in obesity) and better state what drives obesity in children and adults (lack of physical activity see: 10.3389/fped.2020.535705; chronic exposure to pollutants in utero and during life see: 10.1111/obr.13552; genetic predisposition see: 10.1038/s41576-021-00414-z,).
Check the abstract for some minor typo glitches.
Comments on the Quality of English LanguageEnglish requires a moderate check. Authors should improve the overall quality of description.
Author Response
Reviewer 3
Dear Reviewer,
Thank you for your comments, which have helped us improve the quality of our manuscript. Below, you will find a point-by-point analysis of your comments along with our responses.
Major criticisms:
The scarce sample numerosity underpowers the analysis. did the authors conduct a sample size analysis to ensure statistical significancy of the sample and then of the analysis? Discuss this aspect eventually reporting similar analysis on the topic and referring to sample size of such studies. I would eventually state the scarce numerosity as a critical limitation of the study.
Thank you for your comment. Regarding the sample size, the statistical analysis of our study group has confirmed the significance of our analysis. We acknowledge that the sample size of our study is one of its limitations. We have specified this information in the text, emphasizing that our results may not be generalizable to the general pediatric population. However, it is important to note that hyperinsulinemia is not a common condition in pediatric age. Therefore, an analysis conducted on 139 children (of whom 95 had a peak insulin level >100 uIU/mL detected during OGTT) is significant and useful, prospectively, for the realization of large-scale prospective studies. Additionally, we have integrated the discussion. We hope that our revisions meet your expectations.
Could a different metabolic regulation occurring at puberty between males and females have impacted the results? Please address this aspect and eventually consider a further analysis separating males and females, to see if the variables considered are still correlated gender-wise.
Thank you for your comment. We have integrated our statistical analysis by conducting subgroup analysis based on the patients' sex (males vs. females) and pubertal stage (using 9 years as the age limit). The results are reported within the manuscript, although we note that the study subgroups are quite small. Subsequent studies may focus on this issue. We appreciate your feedback, which has allowed us to broaden the study findings.
Importantly, it's unclear how the authors assesses hyperinsulinemia (which should be the reference parameter to correlate with other variables); The authors state: "In the absence of a common guideline for interpreting insulin levels during OGTT we considered a peak insulin level >100 uIU/mL to be pathological, as commonly done in our center". Please: A) reference this passage (if possible) or further discuss, with citing appropriate literature, the rationale behind this methodological choice, which sounds arbitrary. B) Clearly state how you defined hyperinsulinemia in the patients considered for the analysis. For example you report in the inclusion criteria as well as in the discussion the phrase " suspected insulinism". Was the insulinism "suspected" or ascertained? Please be clear in the description of methodologies ant throughout the text. This aspect should be addressed and you should clearly state in the text if you are correlating clear hyperinssulinemima or suspected insulinemia The choice of parameters to define hyperinsulinemia is critical for correctness of the analysis and warrants reliability of results gained.
Thank you for your observation. Currently, the OGTT represents the main method for assessing insulin peak, both in pediatric and adult populations. However, there is significant variability in the interpretation of insulin peaks during OGTT, with variable cut-offs based on the type of guidelines considered or the experience of individual centers. A defined cut-off value in the pediatric population has not yet been established, and there are several possible cut-offs, including:
- The sum of insulin measurements at different sampling times during the OGTT > or <2083.5 pmol/L (300 μU/mL).
- An insulin peak ≥ 1041.75 pmol/L
- A blood insulin value ≥ 520.88 pmol/L (75 μU/mL) when sampled 120 minutes after glucose loading.
- An insulin peak above 100 uIU/mL.
We have provided better clarification of this information within the text, along with a reference for the insulin peak used as a cutoff. Below are some references we have incorporated into our manuscript:
De Sanctis, V.; Soliman, A.; Daar, S.; Tzoulis, P.; Di Maio, S.; Kattamis, C. Oral glucose tolerance test: How to maximize its diagnostic value in children and adolescents. Acta Biomed. 2022, 93, e2022318.
Sahin, N.M.; Kinik, S.T.; Tekindal, M.A. OGTT results in obese adolescents with normal HOMA-IR values. J. Pediatr. Endocrinol. Metab. 2013, 26, 285–291.
Thank you for your valuable input, and we trust that these references strengthen the discussion in our manuscript.
Regarding the observation on hyperinsulinemia, in 139 patients (100%), there is suspected hyperinsulinemia, while it was confirmed in 95 patients based on the insulin peak greater than 100 uIU/mL during OGTT. We have performed correlation analyses on patients with confirmed hyperinsulinemia, and our results refer to that patient group. We have clarified this concept further in the text, and we hope that the changes made will meet your expectations.
The authors are required to critically and clearly discuss the limitations of their work (some of them have been suggested in these comments).
We have incorporated the limitations of our study into the discussion section, as suggested during the review process.
State the type of study you are conducting (cross-sectional assessing correlations etc). For example, in the conclusion you state "Multivariate analysis highlighted that IGF-1 level higher than 203 ng/mL and HOMA index higher than 6.2 are respectively associated with a 9- and 18-times higher risk of hyperinsulinism". The association "marker-risk of something" can be assessed in perspective studies, in which patients are followed for a certain span o time. Pleasese address this aspect and clearly state your conclusion to convey the core message of your paper in order to make it useful for reseachers and clinicians.
Thank you for your observation. It was indeed an error during the translation of the manuscript. In fact, our study is a retrospective monocentric analisys. We have amended the term used.
The conclusions can be improved may be suggesting what you found positively associated with hyperinsulineima. Also, suggest the usefulness and relevancy of such analysis from a clinical standpoint and in the context of exsiting scenario (wherever appropriate, may be also in the discussion).
Thank you once again for your valuable comments, which have helped us enhance the quality of our article. We have revised the conclusion, additionally incorporating the potential clinical implications of our study. We hope that the changes made meet your expectations.
Minor criticisms
In the introduction and eventually in the discussion, the authors are asked to provide more information on the relationship obesity-hyperinsulinemia (what upregulates insulin in obesity) and better state what drives obesity in children and adults (lack of physical activity see: 10.3389/fped.2020.535705; chronic exposure to pollutants in utero and during life see: 10.1111/obr.13552; genetic predisposition see: 10.1038/s41576-021-00414-z,).
Check the abstract for some minor typo glitches.
English requires a moderate check. Authors should improve the overall quality of description.
Thank you for your observation. We have made extensive revisions to our manuscript, incorporating the references you suggested. We trust that these revisions address your concerns adequately.
Round 2
Reviewer 3 Report
Comments and Suggestions for Authors
Some issues have been addressed (superficially), while other not. I am still not convinced by the methodology adopted.
Comments on the Quality of English LanguageEnglish requires a minor check.